# Vasoactive-Inotropic Score as an Early Predictor of Mortality in Adult Patients with Sepsis

**DOI:** 10.3390/jcm10030495

**Published:** 2021-01-31

**Authors:** Juhyun Song, Hanjin Cho, Dae Won Park, Sungwoo Moon, Joo Yeong Kim, Sejoong Ahn, Seong-geun Lee, Jonghak Park

**Affiliations:** 1Department of Emergency Medicine, Korea University Ansan Hospital, Ansan-si 15355, Gyeonggi-do, Korea; songcap97@hotmail.com (J.S.); chohj327@gmail.com (H.C.); yg9912@korea.ac.kr (S.M.); blj01he@naver.com (J.Y.K.); sejoongahn@naver.com (S.A.); whoa891@gmail.com (S.-g.L.); 2Division of Infectious Diseases, Department of Internal Medicine, Korea University Ansan Hospital, Ansan-si 15355, Gyeonggi-do, Korea; pugae1@daum.net; 3National Emergency Medical Center, National Medical Center, Seoul 04564, Korea

**Keywords:** vasopressors, inotropes, sepsis, septic shock, mortality

## Abstract

Vasoactive and inotropic medications are essential for sepsis management; however, the association between the maximum Vasoactive-Inotropic score (VISmax) and clinical outcomes is unknown in adult patients with sepsis. We investigated the VISmax as a predictor for mortality among such patients in the emergency department (ED) and compared its prognostic value with that of the sequential organ failure assessment (SOFA) score. This single-center retrospective study included 910 patients diagnosed with sepsis between January 2016 and March 2020. We calculated the VISmax using the highest doses of vasopressors and inotropes administered during the first 6 h on ED admission and categorized it as 0–5, 6–15, 16–30, 31–45, and >45 points. The primary outcome was 30-day mortality. VISmax for 30-day mortality was significantly higher in non-survivors than in survivors. The mortality rates in the five VISmax groups were 17.2%, 20.8%, 33.3%, 54.6%, and 70.0%, respectively. The optimal cut-off value of VISmax to predict 30-day mortality was 31. VISmax had better prognostic value than the cardiovascular component of the SOFA score and initial lactate levels. VISmax was comparable to the APACHE II score in predicting 30-day mortality. Multivariable analysis showed that VISmax 16–30, 31–45, and >45 were independent risk factors for 30-day mortality. VISmax in ED could help clinicians to identify sepsis patients with poor prognosis.

## 1. Introduction

Sepsis is a leading cause of mortality and morbidity in critically ill patients [1,2,3,4]. Despite significant advances in sepsis management, mortality associated with sepsis and septic shock is high [5,6]. Early detection and appropriate management in the initial hours of hospitalization can improve clinical outcomes of sepsis. The Surviving Sepsis Campaign (SSC) guidelines recommend an initial target mean arterial pressure (MAP) of 65 mmHg in patients with septic shock requiring vasopressors [7]. They suggest the use of vasopressors in patients with sepsis and hypotension not responding to initial fluid resuscitation. Based on a physiologic rationale, the SSC guidelines suggest using dobutamine in patients with evidence of persistent tissue hypoperfusion despite adequate fluid resuscitation and the use of vasopressors [7].

Gaies et al. proposed the Vasoactive-Inotropic score (VIS) to accurately describe cardiovascular dysfunction and predict outcomes in infants after cardiopulmonary bypass [8]. The VIS quantifies the amount of cardiovascular support and includes dopamine, dobutamine, epinephrine, milrinone, vasopressin, and norepinephrine. In previous studies, it correlated with worse outcomes in infants undergoing cardiac surgery [8,9,10,11]. Other studies have suggested that the VIS is associated with clinical outcomes in pediatric sepsis [12,13]. However, to our knowledge, no study has employed the maximum VIS (VISmax) calculated in the emergency department (ED) to predict clinical outcomes in adult patients with sepsis and septic shock.

In this study, we aimed to investigate the association between VISmax and short-term mortality in patients diagnosed with sepsis in the ED according to the latest Sepsis-3 definitions. We hypothesized that VISmax could predict short-term mortality in adult patients with sepsis. We compared its predictive value with that of the sequential organ failure assessment (SOFA) score, the cardiovascular component of the SOFA score, and initial lactate levels.

## 2. Materials and Methods

### 2.1. Study Design and Population

This single-center retrospective cohort study assessed adult patients aged ≥18 years diagnosed with sepsis in the ED at Korea University Ansan Hospital from 1 January 2016 to 31 March 2020. Patients were enrolled if they presented with the presence or suspicion of infection and an increase in SOFA score of ≥2 points. All the enrolled patients with sepsis were carefully selected and reviewed by two infectious diseases (ID) experts. If the patients had baseline (previous) SOFA scores, we used the standard of an increase in SOFA score by at least 2. If the patients had no previous SOFA score, two ID experts reviewed the medical records with laboratory data and determined the association between the current infection and the present SOFA score. We managed the enrolled patients in accordance with the 2016 SSC guidelines [7]. The exclusion criteria were as follows: (a) age <18 years; (b) duration of ED stay <6 h; (c) non-compliance with any items of the 2016 SSC guidelines; (d) missing data (clinical variables or survival outcomes); and (e) extracorporeal membrane oxygenation (ECMO) prior to initiation of vasopressors or inotropes. Eligible patients were divided into the following five groups based on the quintile of VISmax previously described by Koponen et al. to investigate the association between VISmax and outcomes after cardiac surgery [14]: 0–5 (group 1), 6–15 (group 2), 16–30 (group 3), 31–45 (group 4), and >45 (group 5) (Figure 1). 

### 2.2. Data Collection

Using data from the sepsis registry collected by the intelligent sepsis management system of our institution [15], we retrospectively retrieved information on baseline characteristics, clinical data, and patient outcomes. Data on demographics, laboratory results, infection sites, disease severity (sepsis or septic shock), interventions (fluid resuscitation and intravenous antibiotics), and use of vasopressors or inotropes within 6 h after ED admission were recorded. 

### 2.3. Definitions

Currently, sepsis is defined as a life-threatening organ dysfunction caused by a dysregulated host response to infection [1]. Septic shock is defined as a subset of sepsis in which profound circulatory, cellular, and metabolic abnormalities pose a greater risk of mortality than sepsis alone [1,2]. The latest Sepsis-3 definitions recommend the use of the quick SOFA (qSOFA) score to find patients with poor prognosis outside the intensive care unit (ICU) [1]. This score uses three criteria—low blood pressure (systolic blood pressure: ≤100 mmHg), high respiratory rate (≥22 breaths/min), and altered mental status (Glasgow coma score: <15)—assigning 1 point for each criterion, with the final score ranging from 0 to 3 points. A positive qSOFA score is defined as the presence of ≥2 qSOFA points near the onset of infection. We used this positive qSOFA score as the inclusion criterion for our study. According to the Sepsis-3 definition, the diagnostic criteria for sepsis include an increase in the SOFA score by ≥2 points due to current infection. Similarly, the criteria for septic shock include vasopressor requirement to maintain a MAP of 65 mmHg and serum lactate level >2 mmol/L despite adequate fluid resuscitation [1,2].

The VISmax was calculated as follows using the maximum dosing rates of vasopressors and inotropes during the first 6 h after ED admission, which were retrieved from the ED information system:
VISmax =dopamine dose (µg/kg/min) +dobutamine dose (µg/kg/min) +100 × epinephrine dose (µg/kg/min) + 10 × milrinone dose (µg/kg/min) +10,000 × vasopressin dose (units/kg/min) +100 × norepinephrine dose (µg/kg/min)

### 2.4. Outcomes

The primary outcome was 30-day mortality. The secondary outcomes were 7-day and 14-day mortality. Historically, sepsis deaths have occurred in a biphasic distribution, with an initial early peak at several days due to inadequate resuscitation, resulting in cardiac and pulmonary failure, and a late peak at several weeks due to persistent organ injury and failure. In the present study, 7-day and 14-day mortality were used to reflect an initial early peak due to inadequate resuscitation, while 30-day mortality was used to reflect a late peak due to persistent organ injury and failure. The clinical outcomes were extracted from electronic health records of the patients.

### 2.5. Statistical Analysis

Based on prior studies, we expected the 30-day all-cause mortality to be 35%. In previous studies, a vasopressor-driven mortality prediction model predicted mortality better than the SOFA score (area under the receiver operating characteristic curve (AUC) = 0.73 vs. 0.65) [3,16]. Our hypothesis was that we would observe similar AUCs in the present study. Assuming 95% power with a two-sided alpha level of 0.05, our study required 521 patients (339 survivors and 182 non-survivors).

To compare baseline characteristics between survivors and non-survivors, continuous variables, presented as median (interquartile range (IQR)) or mean ± standard deviation, were compared using Student’s t-test or Mann–Whitney test according to the distribution. Data were tested for normality using the Kolmogorov–Smirnov and Shapiro–Wilk tests. Categorical variables, presented as numbers and percentages, were compared using the chi-square test or Fisher’s exact test. To compare clinical characteristics and outcomes (short-term mortalities) between the five VIS groups, continuous variables, presented as median IQR or mean ± standard deviation, were compared using analysis of variance or the Kruskal–Wallis test. Categorical variables, presented as numbers and percentages, were compared using the chi-squared test or Fisher’s exact test. Pairwise comparisons were performed separately for each pair of VIS groups. The Bonferroni method was used to adjust the *p*-value in post hoc analysis. 

Univariable and multivariable Cox proportional hazard models were used to analyze the association between candidate predictive factors and outcomes. Variables with a *p*-value of <0.2 in univariate analysis were further analyzed in the multivariate model, and a stepwise backward elimination method was used to select predictors of 30-day mortality. Data are presented as hazard ratios and 95% confidence intervals (CIs). The predictive value of VISmax, the SOFA score, the acute physiology and chronic health evaluation (APACHE) II score, the cardiovascular component of SOFA score, and initial lactate levels for mortality were assessed using AUCs. We performed a pairwise comparison between the receiver operating characteristic (ROC) curves using the nonparametric Delong method [17]. Cumulative survival curves of individual VIS groups were generated using Kaplan–Meier curve analysis and compared using the log-rank test.

All analyses were performed using MedCalc for Windows, version 19.1.6 (MedCalc Software, Mariakerke, Belgium) and SPSS version 23.0 (IBM, Armonk, NY, USA). A *p*-value of <0.05 was considered statistically significant.

## 3. Results

During the study period, we screened 1558 consecutive patients with sepsis. After excluding 648 patients, 910 patients were finally enrolled in the analysis (Figure 1). Of them, 488 (53.6%) received vasopressors or inotropes within 6 h after ED admission. The VIS groups were 0–5 (*n* = 424), 6–15 (*n* = 72), 16–30 (*n* = 174), 31–45 (*n* = 130), and >45 (*n* = 110) (Figure 1).

The median age of the enrolled patients was 76 years (IQR: 65–82 years), and 518 (56.9%) of the patients were male (Table 1). The median VISmax was 9.0 (IQR: 0.0–36.0). The VISmax (median (IQR)) of non-survivors was significantly higher than that of survivors (36.0 (5.8–54.0) vs. 0.0 (0.0–18.0); *p* < 0.001). There were significant differences in age, occurrence of septic shock, SOFA score, initial lactate levels, procalcitonin levels, and C-reactive protein (CRP) levels between non-survivors and survivors. Compared with survivors, non-survivors were older, more frequently experienced septic shock, had higher SOFA scores, and had higher lactate, procalcitonin, and CRP levels.

Table 2 shows the clinical characteristics of the patients according to the VIS. Patient characteristics, including age, sex, Charlson comorbidity index, septic shock, infection sites, and time to first antibiotics, were not significantly different among the five VIS groups. Overall, patients with a higher VIS had increased 3-h and 6-h fluid administration, higher initial lactate levels, and higher SOFA scores than those with a lower VIS.

Overall, 294 (32.3%) patients died within 30 days of ED admission. The 7-, 14-, and 30-day mortality was 179 (19.7%), 237 (26.0), and 294 (32.3%), respectively (Table 3). Patients with a higher VIS had higher risk of 7-, 14-, and 30-day mortality. The chi-squared test revealed a significant difference in 30-day mortality among the five groups (*p* < 0.001). We conducted a pairwise comparison of 30-day mortality between each VIS group as follows: group 1 vs. group 2, *p* = 0.458; group 1 vs. group 3, *p* < 0.001; group 1 vs. group 4, *p* < 0.001; group 1 vs. group 5, *p* < 0.001; group 2 vs. group 3, *p* = 0.05; group 2 vs. group 4, *p* < 0.001; group 2 vs. group 5, *p* < 0.001; group 3 vs. group 4, *p* < 0.001; group 3 vs. group 5, *p* < 0.001; and group 4 vs. group 5, *p* = 0.015. Finally, after Bonferroni correction (0.05/10 = 0.005), seven of the 10 pairwise comparisons showed significant differences (*p* < 0.005). 

In the univariable Cox proportional hazards model, VIS 16–30, VIS 31–45, VIS > 45, age, SOFA scores, septic shock, initial lactate levels, CRP levels, procalcitonin levels, fluid at 3 h, and fluid at 6 h appeared to be related with a *p*-value of <0.2 (Table 4). In the multivariable model including all these variables, VIS 16–30, VIS 31–45, and VIS > 45 were independent risk factors for 30-day mortality with VIS 0–5 as the reference group. However, there was no difference in mortality between the group 1 (VIS 0–5) and group 2 (VIS 6–15). Age, SOFA scores, initial lactate levels, fluid in 3 h, and fluid in 6 h were associated with 30-day mortality in the multivariable Cox proportional hazards model.

Kaplan–Meier survival curves of each VIS group are presented in Figure 2. Group 1 did not significantly differ from group 2 (log-rank test: *p* = 0.437). Groups 4 and 5 differed from each other (log-rank test: *p* < 0.001) and from Groups 1–3 (log-rank test: all *p* < 0.001). The two highest VISmax groups showed an increased risk of mortality during the first 15 days, which continued to gradually increase up to 30 days.

We performed ROC curves analysis with VISmax, the APACHE II score, the SOFA score, the cardiovascular component of SOFA score, and initial lactate levels to predict 30-day mortality. We performed pairwise comparisons between the variables. The discrimination power of VISmax for 30-day mortality was comparable to that of the SOFA score (AUC = 0.724; 95% CI: 0.694–0.753 vs. AUC = 0.734; 95% CI: 0.704–0.736; *p* = 0.518). The discrimination power of VISmax for 30-day mortality was comparable to that of the APACHE II score (AUC = 0.721; 95% CI: 0.690–0.748; *p* = 0.632). VISmax had better discrimination power for 30-day mortality than the cardiovascular component of the SOFA score (AUC = 0.659; 95% CI: 0.628–0.690; *p* < 0.001) and initial lactate levels (AUC = 0.655; 95% CI: 0.623–0.686; *p* = 0.001). The optimal cut-off value of VISmax to predict 30-day mortality was 31 (sensitivity: 52.7%, specificity: 83.1%). In addition, we presented 30-day mortality for the six different groups generated by the two VISmax categories and the three lactate groups in Table 5.

We performed subgroup analysis including only septic shock patients (*n* = 410). We performed ROC curves analysis with VISmax, the APACHE II score, the SOFA score, the cardiovascular component of SOFA score, and initial lactate levels to predict 30-day mortality. We performed pairwise comparisons between the variables. The discrimination power of VISmax for 30-day mortality was comparable to that of the SOFA score (AUC = 0.703; 95% CI: 0.655–0.748 vs. AUC = 0.702; 95% CI: 0.654–0.747; *p* = 0.974). The discrimination power of VISmax for 30-day mortality was comparable to that of the APACHE II score (AUC = 0.675; 95% CI: 0.627–0.721; *p* = 0.420). VISmax had better discrimination power for 30-day mortality than the cardiovascular component of the SOFA score (AUC = 0.532; 95% CI: 0.483–0.582; *p* < 0.001) and initial lactate levels (AUC = 0.630; 95% CI: 0.581–0.677; *p* = 0.019). The optimal cut-off value of VISmax to predict 30-day mortality was 27 (sensitivity: 67.7%, specificity: 65.2%) in patients with septic shock. In addition, we performed the multivariable Cox proportional hazards model with only septic shock patients. Similar to the multivariable model with overall sepsis patients, VIS 16–30, VIS 31–45, and VIS >45 were independent risk factors for 30-day mortality with VIS 0–5 as the reference group (all *p* < 0.001).

## 4. Discussion

To the best of our knowledge, this is the first study to assess the prognostic value of VISmax calculated in the ED in patients with sepsis diagnosed according to the Sepsis-3 definitions. Although VISmax showed a prognostic value comparable to the SOFA score and APACHE II score, single use of VISmax had limited predictive value for 30-day mortality.

The latest Sepsis-3 definitions incorporate the SOFA score to assess organ dysfunction and defining sepsis [1]. However, due to advances in critical care, the cardiovascular component of the SOFA score does not accurately reflect the current practice, including the use of various vasoactive and inotropic medications [3]. A previous study showed that a modified cardiovascular component of the SOFA score, which includes serum lactate levels, shock index, and all vasopressors, predicted ICU mortality better than the original cardiovascular component of the SOFA score in adult patients admitted to the ICU [16]. This emphasizes the importance of cardiovascular integrity in the prognostication of critically ill patients. The use of high-dose vasoactive medications would worsen the prognosis of sepsis-related cardiovascular dysfunction [18,19,20]. Consistent with the results of previous studies, our results showed that a high VIS was associated with worse outcomes in patients with sepsis. Furthermore, our study showed that the VIS was superior to the cardiovascular component of the SOFA score and was comparable to the SOFA score in predicting short-term mortality.

Two studies have reported an association between the VIS and outcomes in pediatric sepsis [12,13]. One was conducted in a resource-poor setting and the other was conducted in a resource-rich setting. Both the ICU-based studies suggested that Vasoactive-Inotropic support after ICU admission is associated with poor outcomes. Contrary to these studies, our ED-based study included adult patients with sepsis and evaluated the prognostic value of the VIS calculated in the ED. Thus, our results might reflect early cardiovascular integrity in adult patients diagnosed with sepsis in the ED. 

The VIS has been validated in infants after cardiac surgery [8,9,10,11]. Previous studies have demonstrated an association between a high VIS and poor outcomes after cardiac surgery in pediatric patients. Since then, other studies have suggested that the VIS predicts mortality and morbidity in child and adolescent patients after cardiothoracic surgery [21,22]. A recent study reported that the VIS was associated with in-hospital mortality in patients with cardiogenic shock who required cardiac critical care [23]. The study divided patients with cardiogenic shock into five groups based on the quintile of VISmax and demonstrated that the higher VIS group had worse outcomes than the lower VIS group, which agrees closely with our results.

Vasoactive and inotropic medications are essential for septic shock management. However, these medications are also associated with adverse cardiovascular effects, such as arrhythmia, cardiac and peripheral ischemia, and hypertension/hypotension, which can be fatal in critically ill patients. The present study demonstrated that a high VIS was significantly associated with worse outcomes. Poor outcomes could be associated with refractory septic shock itself or adverse cardiovascular effects of these medications, although there is no certainty about their primary cause. We postulate that the former could be the main cause for poor outcomes. In our study, patients were managed at the hospital according to the SSC guidelines and ED physicians tried to minimize unnecessary use of vasopressors and inotropes to avoid adverse effects.

A recent study including only septic shock used three scoring systems to quantify overall peak vasoactive medication requirements: (1) norepinephrine equivalents, (2) Vasoactive-Inotropic score (VIS) and (3) cumulative vasopressor index [3]. They developed new model by combining mechanical ventilation, APACHE-III, vasopressors, inotropes, and Charlson comorbidity index. The incorporation of quantitative vasopressor usage into a new model was superior to the APACHE-III and SOFA scores in its ability to predict 28-day mortality. However, the discrimination power of the model was not excellent (AUC = 0.73) and was similar to that of the VISmax in our study (AUC = 0.724).

Norepinephrine should be used in patients with sepsis-induced hypotension that does not respond to initial fluid resuscitation [7,24,25,26]. It is the first-line vasopressor in septic shock and associated with fewer adverse effects and lower mortality than dopamine [25,26]. Currently, dopamine is not considered a first-line treatment in patients with sepsis or septic shock and hypotension [7]. Thus, we limited its use to patients with sepsis who had a low risk of tachyarrhythmia or who had bradycardia. In accordance with the SSC guidelines and recent studies, we used norepinephrine as a first-line vasopressor among patients with sepsis and hypotension not responding to fluid therapy. We rarely used dobutamine because its use was restricted to patients with sepsis and myocardial dysfunction [24,26].

Post-cardiac arrest syndrome (PCAS) is a “sepsis-like syndrome,” displaying a similar immunologic profile as sepsis [27]. According to a recent study, the VIS within the first 24 h on admission could predict in-hospital mortality in out-of-hospital cardiac arrest patients admitted to the ICU [28]. The 24-h peak VIS showed an AUC of 0.762 (95% CI: 0.690–0.852). Similar to the PCAS study, our showed that the VIS could be a valuable scoring system for predicting 30-day mortality in adult patients with sepsis and septic shock.

To our knowledge, this was the first study to investigate the association between VISmax in ED and mortality. Because our study showed that early VIS max was associated with poor prognosis, higher VISmax in sepsis patients can make ED physicians to pay close attention to such patients, to arrange ICU immediately, and to try more aggressive resuscitation including steroid or other emerging therapies. Further clinical trials are needed to confirm the clinical value of VISmax in sepsis patients. 

This study has several limitations. First, the retrospective design of our study is vulnerable to selection bias, confounding the results. Second, despite overall compliance with the SSC guidelines, the choice of vasoactive and inotropic medications, strategy of initial fluid resuscitation, and target arterial pressure were at the discretion of the ED physicians, possibly affecting the VISmax and clinical outcomes. Third, we focused on early VISmax assessed in the ED; hence, we may have inevitably overlooked the prognostic value of late VISmax. Fourth, our study did not investigate the effect of each vasoactive or inotropic agent on clinical outcome in sepsis patients. Fifth, VISmax alone had limited value in predicting mortality (AUC <0.8). Finally, this was a single-center study conducted in a tertiary care hospital, limiting the generalization of our results.

## 5. Conclusions

VISmax during the first 6 h after ED admission was associated with increased 30-day mortality in adult patients with sepsis diagnosed using Sepsis-3 definitions. Specifically, VISmax was superior to the cardiovascular component of the SOFA score and initial lactate levels and comparable to the APACHE II score in its ability to predict 30-day mortality. VISmax in ED could help clinicians to identify sepsis patients with poor prognosis.

## Figures and Tables

**Figure 1 jcm-10-00495-f001:**
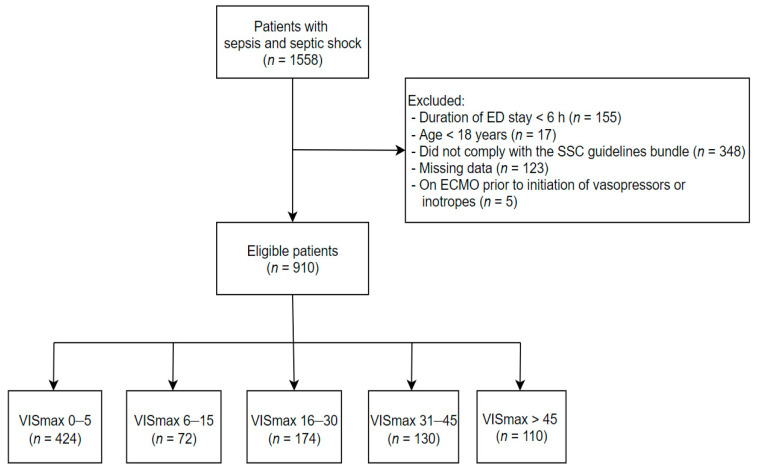
Flowchart of the study population. ED: emergency department; SSC: Surviving Sepsis Campaign; ECMO: extracorporeal membrane oxygenation; VIS: Vasoactive-Inotropic score.

**Figure 2 jcm-10-00495-f002:**
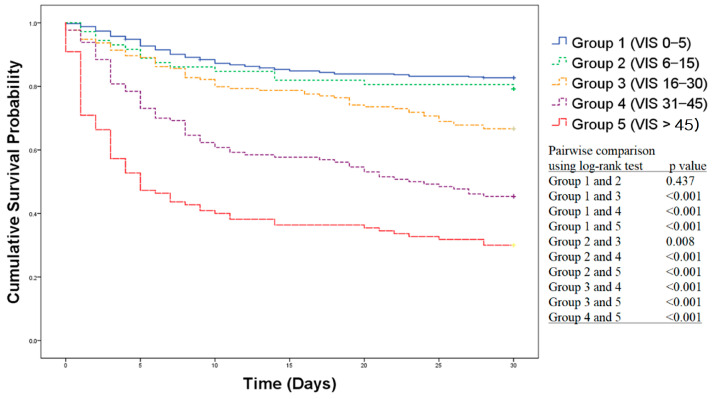
Kaplan–Meier survival curve of each VIS group. VIS: Vasoactive-Inotropic score.

**Table 1 jcm-10-00495-t001:** Baseline patient characteristics according to 30-day mortality.

Variables	All Patients (*n* = 910)	30-Day Mortality		*p*-Value
		Non-Survivors (*n* = 294)	Survivors (*n* = 616)	
Median age, years (IQR)	76 (65–82)	78 (69–84)	74 (63–81)	<0.001
Male, *n* (%)	518 (56.9)	168 (57.1)	350 (56.8)	0.926
Median CCI (IQR)	4 (3–5)	5 (4–6)	4 (3–5)	0.158
Septic shock, *n* (%)	410 (45.1)	201 (68.4)	209 (33.9)	<0.001
VISmax, median (IQR)	9.0 (0.0–36.0)	36.0 (5.8–54.0)	0.0 (0.0–18.0)	<0.001
NEmax, median (IQR), µg/kg/min	0.1 (0.0–0.3)	0.3 (0.1–0.4)	0.0 (0.0–0.2)	<0.001
Infection sites, *n* (%)				
Respiratory	573 (63)	183 (62)	390 (63)	0.691
Genitourinary	215 (24)	73 (25)	142 (23)	0.348
Gastrointestinal	93 (10)	31 (11)	62 (10)	
Others	87 (10)	28 (10)	59 (10)	
Median time to first antibiotics, min (IQR)	120 (71–196)	122 (70–203)	123 (72–205)	0.427
Median fluid volume, mL (IQR)				
within 3 h	1800 (1275–2200)	1600 (1300–2000)	1800 (1200–2400)	0.044
within 6 h	2600 (2000–3300)	2800 (2300–3225)	2600 (1800–3300)	0.045
SOFA score, median (IQR)	8 (6–11)	10 (8–12)	7 (5–9)	<0.001
APACHE II score, median (IQR)	20 (15–25)	23 (18–29)	18 (14–23)	<0.001
Lactate, mmol/L	2.9 (1.8–5.4)	4.3 (2.3–7.8)	2.5 (1.6–4.5)	<0.001
Procalcitonin, ng/mL	1.6 (0.4–11.7)	2.4 (0.6–12.1)	1.3 (0.3–11.5)	0.002
CRP, mg/dL	10.2 (4.6–18.5)	11.9 (6.0–20.2)	9.4 (4.0–18.0)	0.003

IQR: interquartile range; CCI: Charlson comorbidity index; VISmax: maximum Vasoactive-Inotropic score; NEmax: maxium dose of norepinephrine; SOFA: sequential organ failure assessment; APACHE: acute physiology and chronic health evaluation; CRP: C-reactive protein.

**Table 2 jcm-10-00495-t002:** Clinical characteristics according to each VIS group.

Variables	VIS 0–5 (*n* = 424)	VIS 6–15 (*n* = 72)	VIS 16–30 (*n* = 174)	VIS 31–45 (*n* = 130)	VIS > 45 (*n* = 110)	*p*-Value
Median age, years (IQR)	76 (66–82)	77 (67–81)	73 (61–82)	76 (65–83)	77 (66–84)	0.336
Male, *n* (%)	234 (55.2)	36 (50.0)	98 (56.3)	84 (64.6)	66 (60.0)	0.238
Median CCI (IQR)	4 (3–5)	4 (3–5)	4 (3–5)	5 (4–6)	5 (4–6)	0.147
Septic shock, *n* (%)	2 (0.5)	59 (81.9)	132 (75.9)	110 (84.6)	107 (97.3)	<0.001
Infection sites, *n* (%)						
Respiratory	267 (63)	46 (64)	108 (62)	82 (63)	70 (64)	0.687
Genitourinary	98 (23)	17 (24)	42 (24)	30 (23)	28 (25)	0.592
Gastrointestinal	42 (10)	8 (11)	17 (10)	14 (11)	12 (11)	
Others	40 (9)	8 (11)	16 (9)	13 (10)	10 (9)	
Median time to first antibiotics, min (IQR)	119 (70–194)	125 (75–205)	123 (72–200)	121 (71–202)	120 (72–199)	0.397
Median fluid volume, mL (IQR)						
within 3 h	1500 (900–1800)	1800 (1500–2200)	2000 (1500–2500)	2000 (1500–2500)	2400 (1800–2500)	<0.001
within 6 h	2200 (1800–2600)	3000 (2300–3300)	3100 (2600–3600)	3200 (2600–3600)	3350 (3000–3800)	<0.001
SOFA score, median (IQR)	6 (4–7)	9 (8–11)	10 (8–12)	10 (9–12)	11 (9–13)	<0.001
APACHE II score, median (IQR)	17 (13–21)	20 (16–24)	22 (17–28)	22 (18–27)	24 (19–29)	<0.001
Lactate, mmol/L	2.2 (1.5–3.9)	2.9 (2.0–4.6)	3.0 (2.0–5.3)	4.3 (2.5–6.4)	7.8 (4.1–11.3)	<0.001
Procalcitonin, ng/mL	0.7 (0.2–4.3)	2.5 (0.5–18.0)	2.8 (0.8–17.5)	5.6 (0.9–26.7)	3.9 (0.6–21.6)	<0.001
CRP, mg/dL	9.1 (3.5–16.8)	12.6 (5.2–20.9)	11.4 (5.2–20.7)	11.6 (6.2–20.4)	9.4 (4.3–16.6)	0.003

IQR: interquartile range; CCI: Charlson comorbidity index; SOFA: sequential organ failure assessment; APACHE: acute physiology and chronic health Evaluation CRP: C-reactive protein; VIS: Vasoactive-Inotropic score.

**Table 3 jcm-10-00495-t003:** Clinical outcomes (short-term mortality) according to each VIS group.

Outcomes	VIS 0–5 (*n* = 424)	VIS 6–15 (*n* = 72)	VIS 16–30 (*n* = 174)	VIS 31–45 (*n* = 130)	VIS > 45 (*n* = 110)	*p*-Value
7-day mortality	42 (9.9)	10 (13.9)	25 (14.4)	40 (30.8)	62 (56.4)	<0.001
14-day mortality	63 (14.9)	12 (16.7)	37 (21.3)	55 (42.3)	70 (63.6)	<0.001
30-day mortality	73 (17.2)	15 (20.8)	58 (33.3)	71 (54.6)	77 (70.0)	<0.001

VIS: Vasoactive-Inotropic score.

**Table 4 jcm-10-00495-t004:** Predictors of 30-day mortality using the Cox proportional hazards model.

	Univariable HR (95% CI)	*p*-Value	Multivariable HR (95% CI)	*p*-Value
VIS group				
VIS 0–5	1 (Reference group)		1 (Reference group)	
VIS 6–15	1.236 (0.709–2.155)	0.454	1.028 (0.525–2.015)	0.936
VIS 16–30	2.060 (1.459–2.908)	<0.001	1.884 (1.159–3.063)	0.011
VIS 31–45	3.975 (2.866–5.514)	<0.001	3.717 (2.305–5.994)	<0.001
VIS > 45	6.934 (5.025–9.567)	<0.001	6.266 (3.624–10.834)	<0.001
Age, years	1.018 (1.009–1.028)	<0.001	1.014 (1.004–1.025)	0.005
Sex				
Male	1 (Reference group)			
Female	0.989 (0.785–1.246)	0.926		
SOFA score	1.221 (1.181–1.263)	<0.001	1.132 (1.075–1.191)	<0.001
APACHE II score	1.187 (1.148–1.229)	<0.001	1.093 (1.036–1.152)	<0.001
Septic shock				
Sepsis	1 (Reference group)			
Septic shock	3.236 (2.530–4.139)	<0.001		
Lactate	1.103 (1.080–1.128)	<0.001	1.069 (1.034–1.106)	<0.001
CRP	1.013 (1.002–1.024)	0.019	1.003 (0.990–1.016)	0.692
Procalcitonin	1.004 (1.000–1.008)	0.065	0.999 (0.995–1.003)	0.684
Fluid in 3 h	0.9999 (0.9997–1.0000)	0.164	0.998 (0.998–0.999)	<0.001
Fluid in 6 h	1.0001 (1.0000–1.0002)	0.119	1.001 (1.000–1.001)	0.001

HR: hazard ratio; VIS: Vasoactive-Inotropic score; SOFA: sequential organ failure assessment; APACHE: acute physiology and chronic health evaluation; CRP: C-reactive protein.

**Table 5 jcm-10-00495-t005:** Thirty-day mortality for the six different groups generated by the two VISmax categories and the three lactate groups.

	Lactate Group (mmol/L)		
VISmax Category	Lactate ≤ 2 Total, *n* (Died *n*/%)	Lactate > 2 to ≤4 Total, *n* (Died *n*/%)	Lactate > 4 Total, *n* (Died *n*/%)
VISmax < 31	262 (43/16.4)	226 (46/20.4)	182 (57/31.3)
VISmax ≥ 31	28 (15/53.6)	57 (34/59.6)	155 (99/63.9)

## Data Availability

The data presented in this study are available on request from the corresponding author. The data are not publicly available due to their containing information that could compromise the privacy of research participants.

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
