# Peer review of "Vasoactive-Inotropic Score as an Early Predictor of Mortality in Adult Patients with Sepsis"

_jcm, 2021, doi:10.3390/jcm10030495_

Round 1
Reviewer 1 Report
The paper has improved slightly.
There is some confusion between SOFA (an indicator of the degree of organ dysfunction) and the qSOFA (an alarm signal to identify patient at risk).
‘Patients were enrolled if they presented with presence or suspicion of 61 infection and a SOFA score of ≥2 points‘: this is wrong
Sepsis is not infection + SOFA >2, but infection with an increase in SOFA by at least 2 (otherwise a mild infection in a patient with e.g. renal failure would be called sepsis!)
Of course there is no reason to indicate SOFA score should be at least 2, if it must increase by 2!
The implications are still not clear: the authors only added ‘VISmax in ED could help clinicians to identify sepsis patients with poor prognosis’, but do not explain what the clinician could do with this information. We all appreciate that the higher the doses of vasoactive agents, the more severe the situation, but nobody uses a score for this. Maybe it is more important for clinical trials?
Author Response
Comments and Suggestions for Authors
1. There is some confusion between SOFA (an indicator of the degree of organ dysfunction) and the qSOFA (an alarm signal to identify patient at risk). ‘Patients were enrolled if they presented with presence or suspicion of infection and a SOFA score of ≥2 points‘: this is wrong. Sepsis is not infection + SOFA >2, but infection with an increase in SOFA by at least 2 (otherwise a mild infection in a patient with e.g. renal failure would be called sepsis!) Of course there is no reason to indicate SOFA score should be at least 2, if it must increase by 2!
Answer: Thank you for your accurate and thoughtful comments. As you indicated, sepsis is clinically diagnosed as infection with an increase in SOFA score by at least 2. Basically, all the enrolled patients with sepsis were carefully selected and reviewed by two infectious diseases (ID) expert. We understand your concern that our description about using SOFA score have made confusion to the readers. If the patients had baseline (previous) SOFA scores, we used the standard of an increase in SOFA score by at least 2. If the patients had no previous SOFA score, two ID experts carefully reviewed the medical records with laboratory data and determined the association between the current infection and the present SOFA score. Therefore, we had definitely excluded patients who had infection but had no increase in SOFA score by at least 2 (i.e. an increase in SOFA score by 0 or 1). According to your comments, we revised our manuscript. We really appreciate your accurate comments and concern again.
2. The implications are still not clear: the authors only added ‘VISmax in ED could help clinicians to identify sepsis patients with poor prognosis’, but do not explain what the clinician could do with this information. We all appreciate that the higher the doses of vasoactive agents, the more severe the situation, but nobody uses a score for this. Maybe it is more important for clinical trials?
Answer: Thank you for your thoughtful comments. Several previous studies including patients with cardiac surgery or pediatric sepsis, or post-cardiac arrest syndrome showed that VIS is associated with clinical outcomes. To our knowledge, this was the first study that investigated the association between VISmax in ED and mortality. Because our study showed that early VIS max was associated with poor prognosis, higher VISmax in sepsis patients can make ED physicians to pay close attention to such patients, to arrange ICU immediately, and to try more aggressive resuscitation including steroid or other emerging therapies. We also agree with your suggestion that further clinical trials are needed to confirm the prognostic value of VISmax in sepsis patients. Thank you for your time and comments.
Reviewer 2 Report
The Authors have addressed all my concerns. No further comments.
Author Response
Comments and Suggestions for Authors
The Authors have addressed all my concerns. No further comments.
Answer: Thank you for your time and helpful comments. Your suggestion definitely improved our manuscript. We really appreciate your contribution to our article.
Reviewer 3 Report
Jcm-1084807
In this clinical study the predictive value of the vasoactive-inotropic score (VIS) was investigated. They found mortality to increase with increased maximum VIS score (VISmax), and to have better predictive value than SOFA-cardiovascular score and initial lactate level.
It is an interesting but not very controversial finding that intensity of the treatment is correlated with outcomes in critical ill patients. The authors basically find that more vasoactive support is associated with increased mortality.
I have some major concern in using the VISmax.
The first is to mix pure vasopressors agents with inotropic agents and agents with both properties into one score is difficult to interpret. The same score could be achieved with very different profiles of vasopressors and inotropes and to equalize these two may be a shortcut. Before this can be done one need to know the individual prognostic properties of different doses of each of these three groups, which probably does not exists.
The second is to use SOFA score as a score to predict mortality. The development of this score was not intended to be used as a prognostic score, it was to describe organ dysfunction. There are no clear cut association between the individual scores in each organ system and between each step within one sub-score. Hence a proper comparison would better be performed with only one of the traditional prognostic scores like SAPS or APACHE, and not SOFA score (I realize you have used the A-II score as well)
Third, most of the ROC analysis are within the same “clinical value), none reaching values > 0.8, and hence not suitable for individual prognostic evaluation. Even with the optimal VISmax score the sensitivity is close to 0,5.
Fourth, I think you should not mix sepsis and septic shock into one group, but analysed them separately. These two clinical conditions are very different.
I also have a more general view of using treatment variables to measure outcome. We use such management in order to treat a specific abnormality, blood-pressure with vasopressors, cardiac output with inotropes, and hypoxia with more oxygen (increase FiO2). The next step to prove that this improvement in a simple measure (MAP) then increase outcome and that this is caused by inotropes ins a short-cut. Reality is more complex. I would have liked to see such elements dealt with in the discussion in a better way
Minor comments
- Page 1, line 39-40. SSC do not recommend dobutamin for hypotension in septic shock, but for hypoperfusion, this is not the same, please correct
- Page 3. line 86-91 qSOFA is not recommended to find or identify patients with sepsis but to find patients (from any disease) with an increased risk of dying in the hospital. Please correct.
- Page 3 line 100, why secondary outcome of 7 and 14 days? Please explain (I could have understood a secondary outcome > 30 days, since having 30 days you have the two other automatically (if you register time of death of course).
- Reference 3 only studies patients with septic shock, and have separate index for vasopressors and vasoactive inotropes, not combined them (see general comment above)
- Page 4, table 1. The dose of NE max (maximum dose of NE in ug/kg/min) must be wrong (10 and 30 ug/kg/min are far too high, the usual dose is often 0,1 to 0,3 ug/kg/min
- Table 4, all included variables included in the multivariate analysis are probably not independent: VIS, SOFA and septic shock are probably closely dependent, please comment and correct.
Author Response
In this clinical study the predictive value of the vasoactive-inotropic score (VIS) was investigated. They found mortality to increase with increased maximum VIS score (VISmax), and to have better predictive value than SOFA-cardiovascular score and initial lactate level.
It is an interesting but not very controversial finding that intensity of the treatment is correlated with outcomes in critical ill patients. The authors basically find that more vasoactive support is associated with increased mortality.
I have some major concern in using the VISmax.
Major concern
1. The first is to mix pure vasopressors agents with inotropic agents and agents with both properties into one score is difficult to interpret. The same score could be achieved with very different profiles of vasopressors and inotropes and to equalize these two may be a shortcut. Before this can be done one need to know the individual prognostic properties of different doses of each of these three groups, which probably does not exist.
Answer: Thank you for your thoughtful comments. As you commented, VISmax includes pure vasopressors, pure inotropes, and agents with both properties. Therefore, VIS cannot reflect the individual properties of various vasoactive and inotropic agents. One of the major limitation is that our study did not investigate the effect of each agent on clinical outcome in sepsis patients. We added this limitation in our discussion. Further studies with larger cohort are needed to validate the prognostic value of each vasoactive or inotropic agent. We really appreciate your accurate comment for our article.
2. The second is to use SOFA score as a score to predict mortality. The development of this score was not intended to be used as a prognostic score, it was to describe organ dysfunction. There are no clear cut association between the individual scores in each organ system and between each step within one sub-score. Hence a proper comparison would better be performed with only one of the traditional prognostic scores like SAPS or APACHE, and not SOFA score (I realize you have used the A-II score as well)
Answer: Thank you for your accurate comments. As you commented, SOFA score was not developed to predict clinical outcome in sepsis patients. According to your suggestion, we compared the prognostic value of VISmax with that of APACHE II score.
3. Third, most of the ROC analysis are within the same “clinical value), none reaching values > 0.8, and hence not suitable for individual prognostic evaluation. Even with the optimal VISmax score the sensitivity is close to 0,5.
Answer: Thank you for your thoughtful comments. As you commented, AUCs of all the variables in ROC analysis were below 0.8, which means that VISmax alone had limited value in predicting mortality. Therefore, we added table including the six different groups generated by the two VISmax categories and the three lactate groups. This might help clinicians to triage the sepsis patients in ED. Nevertheless, AUC <0.8 of VISmax is another limitation of the present study.
4. Fourth, I think you should not mix sepsis and septic shock into one group, but analysed them separately. These two clinical conditions are very different.
Answer: Thank you for your comments. As you commented, sepsis and septic shock have different characteristics and severity. Mortality is definitely higher in septic shock. According to your suggestions, we performed ROC curve analysis and Cox proportional hazards model including only septic shock patients.
I also have a more general view of using treatment variables to measure outcome. We use such management in order to treat a specific abnormality, blood-pressure with vasopressors, cardiac output with inotropes, and hypoxia with more oxygen (increase FiO2). The next step to prove that this improvement in a simple measure (MAP) then increase outcome and that this is caused by inotropes ins a short-cut. Reality is more complex. I would have liked to see such elements dealt with in the discussion in a better way
Minor comments
1. Page 1, line 39-40. SSC do not recommend dobutamine for hypotension in septic shock, but for hypoperfusion, this is not the same, please correct
Answer: Thank you for your thoughtful comments. According to your suggestions, we revised our manuscript. We described the related contents in accordance with the SSC 2016 guidelines.
2. Page 3. line 86-91 qSOFA is not recommended to find or identify patients with sepsis but to find patients (from any disease) with an increased risk of dying in the hospital. Please correct.
Answer: Thank you for your accurate comments. Because qSOFA has only limited sensitivity for detecting sepsis, it is not recommended to find or identify overall sepsis patients. I am sorry that our article included incorrect description about qSOFA. According to your comments, we revised our manuscript. “The latest Sepsis-3 definitions recommend the use of the qSOFA score to find patients with poor prognosis outside the intensive care unit.”
3. Page 3 line 100, why secondary outcome of 7 and 14 days? Please explain (I could have understood a secondary outcome > 30 days, since having 30 days you have the two other automatically (if you register time of death of course).
Answer: Thank you for your thoughtful comments. According to conventional sepsis mortality distribution, sepsis deaths have occurred in a biphasic distribution, with an initial early peak at several days due to inadequate resuscitation, resulting in cardiac and pulmonary failure, and a late peak at several weeks due to persistent organ injury and failure. In the present study, 7-day and 14-day mortality were used to reflect an initial early peak due to inadequate resuscitation, while 30-day mortality was used to reflect a late peak due to persistent organ injury and failure. In Kaplan-Meier survival curve analysis (figure 2), the patients showed an increased risk of mortality during the first 15 days, which continued to gradually increase up to 30 days. Because we investigated only 30 days on ED presentation, our results might have not fully reflected the late peak at several weeks. Further follow-up study is needed to see the pattern of the late peak in the study population.
4. Reference 3 only studies patients with septic shock, and have separate index for vasopressors and vasoactive inotropes, not combined them (see general comment above)
Answer: Thank you for your thoughtful comments. As you commented, Reference 3 only investigated patients with septic shock. They used three scoring systems to quantify overall peak vasoactive medication requirements: (1) norepinephrine equivalents (NEE), (2) vasoactive inotropic score (VIS) and (3) cumulative vasopressor index (CVI). They developed new model by combining Mechanical ventilation, APACHE, Vasopressors, Inotropes, and Charlson comorbidity index. The incorporation of quantitative vasopressor usage into a new model was superior to the APACHE-III and SOFA scores in its ability to predict 28-day mortality. However, the discrimination power of the model was not excellent (AUC = 0.73) and was similar to that of the VISmax in our study (AUC = 0.724).
5. Page 4, table 1. The dose of NE max (maximum dose of NE in ug/kg/min) must be wrong (10 and 30 ug/kg/min are far too high, the usual dose is often 0,1 to 0,3 ug/kg/min
Answer: Thank you for your accurate comments. As you commented, the dose of NEmax was definitely wrong. Therefore, we corrected the wrong values in Table 1. (10-->0.1 & 30-->0.3 µg/kg/min).
6. Table 4, all included variables included in the multivariate analysis are probably not independent: VIS, SOFA and septic shock are probably closely dependent, please comment and correct.
Answer: Thank you for your thoughtful comments. Of course, we understand your concern about the correlation between the variables (VISmax, SOFA score, APACHE II score, septic shock, and lactate). We previously performed correlation analysis between the variables. Overall, there were positive correlations between the variables (rho = 0.526-0.583, all p <0.001). Therefore, to eliminate redundant variables with multicollinearity, we performed multivariable Cox proportional hazard model with “backward LR (likelihood ratio) elimination method”. After this analysis, “septic shock” was finally excluded in the model. Except “septic shock”, other variables (VISmax, SOFA score, APACHE II score, and lactate) remained in the final model.
Round 2
Reviewer 3 Report
I thank the authors for their corrections and additional comments. No further changes required.
This manuscript is a resubmission of an earlier submission. The following is a list of the peer review reports and author responses from that submission.
Round 1
Reviewer 1 Report
COMMENTS FOR THE AUTHOR:
Thank you for the opportunity to review the manuscript. The authors report a single-center retrospective study with 910 patents recruited from the emergency department diagnosed with sepsis and septic shock.
The authors proposed an interesting and complete analysis in order to analyze the association between the maximum vasoactive–inotropic score (VISmax) and mortality. They conclude that VISmax predict 30-day mortality in adults patients with sepsis and septic shock. It is a well-designed study with interesting results, but there are some areas of concern, as follows:
- Methods: Please details Septic Shock definition.
- Results:
- Table 1: In addition to the time of introduction of antibiotic treatment, it would be interesting to include whether antibiotic treatment has been successful, according to guidelines or according to microorganism.
- Table 1: Check the number of patients with Septic Shock. (Total: 500, non-survivors: 201, survivors: 209)
- Table 4: Have you performed a correlation analysis to exclude redundant variables? If the authors have used the definition of sepsis and septic shock according to SEPSIS 3, Lactate would be included in the septic shock definition.
- Have the authors considered studying other complications such as mediastinitis, stroke, acute kidney injury, or myocardial infarction?
- Did the authors compare the VISmax score with other widely used scoring systems such as APACHE II or SAPS II?
- Please include statistical significance markers in figures.
Author Response
REVIEWER #1
Thank you for the opportunity to review the manuscript. The authors report a single-center retrospective study with 910 patents recruited from the emergency department diagnosed with sepsis and septic shock.
The authors proposed an interesting and complete analysis in order to analyze the association between the maximum vasoactive–inotropic score (VISmax) and mortality. They conclude that VISmax predict 30-day mortality in adult patients with sepsis and septic shock. It is a well-designed study with interesting results, but there are some areas of concern, as follows:
Answer: Thank you for your time and helpful comments. We tried to do our best to reflect your comments and revised our manuscript according to your suggestions.
Methods: Please details Septic Shock definition.
Answer: Thank you for your helpful comments. We described the details the definition and diagnostic criteria of Sepsis in 2.3. Definitions of Materials and Methods.
Results:
Table 1: In addition to the time of introduction of antibiotic treatment, it would be interesting to include whether antibiotic treatment has been successful, according to guidelines or according to microorganism.
Answer: Thank you for your helpful comments. Surviving Sepsis Campaign guidelines 2016 recommend IV broad-spectrum antibiotics within 1 hour upon the recognition of sepsis development. First, as you suggested, we investigated the association between timeliness of antibiotics (within 1 hour vs over 1 hour) and 30-day mortality (dead or alive). However, timeliness of antibiotics was not associated with 30-day mortality (p = 0.493). Second, we investigated the appropriate choice of antibiotics with the help of the infectious diseases (ID) specialist working in our institution. If IV broad-spectrum antibiotics were used empirically according to the suspected sites of infection, the ID expert considered those cases as appropriate choice of antibiotics. Among overall 910 patients, 850 patients (93.4%) had appropriate choice of antibiotics. We also analyzed the association between appropriateness of antibiotics choice (appropriate vs inappropriate) and 30-day mortality. The appropriateness of antibiotics choice was not associated with 30-day mortality in the present study (p = 0.218). We postulate that further studies with larger cohort are needed to evaluate the association between appropriate antibiotics use and clinical outcomes.
Table 1: Check the number of patients with Septic Shock. (Total: 500, non-survivors: 201, survivors: 209)
Answer: Thank you for your accurate comments. There was an error in calculating the total number of patients with septic shock. The correct number of patients with septic shock (201 + 209 = 410) is presented in Table 1.
Table 4: Have you performed a correlation analysis to exclude redundant variables? If the authors have used the definition of sepsis and septic shock according to SEPSIS 3, Lactate would be included in the septic shock definition.
Answer: Thank you for your comments and concern about the multicollinearity. We consulted with our statistical expert again regarding your suggestions. As you commented, we had performed correlation analysis between lactate and septic shock. There was a positive correlation between the two variables (rho = 0.536, p <0.001). To eliminate redundant variables, we performed multivariate Cox proportional hazard model with “backward LR (likelihood ratio) elimination method”. After this analysis, “septic shock” was finally excluded in the model. We totally agree with your comments about redundant variables. In table 4, we excluded redundant variable (septic shock) in multivariate analysis.
Have the authors considered studying other complications such as mediastinitis, stroke, acute kidney injury, or myocardial infarction?
Answer: Thank you for your helpful comments. As you suggested, we investigated complications possibly related to the use of vasopressors or inotropes. After the use of vasoactive and inotropic agents, we observed 23 patients with acute kidney injury,11 patients with hypertensive emergency, 7 patients with arrhythmia, 4 patients with acute myocardial infarction during hospital admission. However, it is uncertain whether such complications were caused directly by the vasoactive and inotropic agents or not.
Did the authors compare the VISmax score with other widely used scoring systems such as APACHE II or SAPS II?
Answer: Thank you for your helpful comments. As you commented, APACHE and SAPS score are widely used as a prognostic score in ICU admitted patients. However, we could not compare the VISmax with APACHE II or SAPS II, because some variables for APACHE II or SAPS II calculation were missing in our registry of sepsis patients.
Please include statistical significance markers in figures.
Answer: Thank you for your comments. According to your suggestions, we included statistical significance markers in figure 2 and 3.
Reviewer 2 Report
The authors retrospectively studied the prognostic value of the VISmax in septic patients in their emergency department (ED).
As expected, the prognostic value was present, although not very good.
General comments
- Why is it important to define prognosis? Obviously not to limit therapy – as an example, a young patient with meningococcemia may require huge doses of cardiovascular agents and be fully treated, like the patient with peritonitis on its way to the OR. Obviously all these patients must be admitted in the ICU, so that it has no implications for patient management. The SOFA score was introduced to describe the severity of organ dysfunction, but not to predict outcome, like APACHE or SAPS scores. And the latter scores are useful for scientific publications rather than to daily application.
- The predictive value is actually very poor (AUROC hardly above 0.75). This should be clearly recognized. The Figure 3 could be omitted. (the Figure 2 is better).
- The monocentric nature of the study is real handicap. As the authors indicate, dopamine should not be used, epinephrine use if limited to extremely severe cases, milrinone is hardly used and the use of vasopressin is highly variable. Which agents were used in these patients? Is the max dose of norepi not an easier information about the severity of the shock state? (that is the information that people use in their communication)
- This study is about septic shock, isn’t it? Why would we use these agents if there is no alteration in tissue perfusion? Speaking about ‘sepsis and septic shock’ is rather confusing.
Other specific comments
- ‘sepsis and septic shock’ do not make sense: septic shock is a subset of sepsis. All patients with septic shock have sepsis. Septic patients may or may not be in shock.
- Line 43: ‘in patients with sepsis and hypotension not responding to initial fluid resuscitation’: isn’t it septic shock (lactate levels are usually increased in these circumstances)?
- Line 62: ‘positive quick SOFA (qSOFA) score, presence or suspicion of infection, and a SOFA score of ≥2 points…’: if they have a positive qSOFA, by definition they must have a SOFA score of at least 2.
- The authors should be emphasize that the AUROC is very poor in all cases (hardly 0.75).
Author Response
REVIEWER #2
Comments and Suggestions for Authors
The authors retrospectively studied the prognostic value of the VISmax in septic patients in their emergency department (ED).
As expected, the prognostic value was present, although not very good.
Answer: Thank you for your time and helpful comments. We tried to do our best to reflect your comments and revised our manuscript according to your suggestions.
Comments and Suggestions for Authors
COMMENTS FOR THE AUTHOR:
General comments
- Why is it important to define prognosis? Obviously not to limit therapy – as an example, a young patient with meningococcemia may require huge doses of cardiovascular agents and be fully treated, like the patient with peritonitis on its way to the OR. Obviously all these patients must be admitted in the ICU, so that it has no implications for patient management. The SOFA score was introduced to describe the severity of organ dysfunction, but not to predict outcome, like APACHE or SAPS scores. And the latter scores are useful for scientific publications rather than to daily application.
Answer: Thank you for your helpful comments. As you commented, we assessed the prognostic value early VISmax calculated in ED and did not evaluate VISmax in ICU. Therefore, that is one of the limitations in the present study. We totally agree with your comments about the role of SOFA score, APACHE, and SAPS scores. APACHE and SAPS score are known to be useful to prognosticate critically ill patients in ICU. Unfortunately, some variables for calculating APACHE and SAPS scores were missing in our registry data. For this reason, we were compelled to compare VISmax with only SOFA score.
- The predictive value is actually very poor (AUROC hardly above 0.75). This should be clearly recognized. The Figure 3 could be omitted. (the Figure 2 is better).
Answer: Thank you for your helpful comments. As you commented, the prognostic value of VISmax proved to be not good, but only fair (AUC = 0.724). Generally, AUCs are classified as follows: 1) >0.9: excellent, 2) 0.8–0.9: good, 3) 0.7–0.8: fair, 4) 0.6–0.7: poor, 5) <0.6: fail. Therefore, we concluded that VISmax had a fair prognostic value comparable to SOFA score. Figure 3 shows the ROC curves of individual variables to predict 30-day mortality. If you really suggest that the Figure 3 would be unnecessary in our article, we will omit the figure
- The monocentric nature of the study is real handicap. As the authors indicate, dopamine should not be used, epinephrine use if limited to extremely severe cases, milrinone is hardly used and the use of vasopressin is highly variable. Which agents were used in these patients? Is the max dose of norepi not an easier information about the severity of the shock state? (that is the information that people use in their communication)
Answer: Thank you for your thoughtful comments. As you commented, the most commonly used vasopressor was norepinephrine (NE) because it is recommended as first line agent for sepsis-induced hypotension. We agree with your point that max dose of Norepinephrine could be a simple and easier way to evaluate the severity of septic shock. However, in a few patients, dopamine or dobutamine was initially used depending on their clinical condition or clinicians’ decision. As you commented, epinephrine and vasopressin was used in refractory septic shock patients. However, milrinone has never been used in our study. Although, VIS was not designed to reflect the use of vasopressors or intropes in sepsis, our study showed that VIS has fair prognostic value in sepsis and septic shock.
- This study is about septic shock, isn’t it? Why would we use these agents if there is no alteration in tissue perfusion? Speaking about ‘sepsis and septic shock’ is rather confusing.
Answer: Thank you for your helpful comments. As you commented, if there is no tissue hypoperfusion, vasopressors might be unnecessary. This study included both sepsis (without shock) and septic shock. To diagnose septic shock, sepsis patients should meet two of the diagnostic criteria 1) vasopressor requirement to maintain a MAP of 65 mmHg and 2) serum lactate level >2 mmol/L despite adequate fluid resuscitation. Therefore, our study included some sepsis (not septic shock) patients with <2 mmol/L of serum lactate levels requiring vasopressors. Of course, our study also included sepsis (without shock) patients who did not received any vasopressor or inotrope. In the present study, the term “sepsis” indicates “sepsis without shock”.
Other specific comments
- ‘sepsis and septic shock’ do not make sense: septic shock is a subset of sepsis. All patients with septic shock have sepsis. Septic patients may or may not be in shock.
Answer: Thank you for your comments. As you suggested, septic shock is a subset of sepsis. However, in our manuscript, the term ‘sepsis and septic shock’ means ‘sepsis without shock and sepsis with shock’. In many previous studies according to the latest Sepsis-3 definitions, the term ‘sepsis and septic shock’ has been widely used. In those articles, ‘sepsis and septic shock’ means ‘sepsis without shock and sepsis with shock’. Similar to these, in many previous studies using Sepsis-2 definitions, the term ‘sepsis, severe sepsis, and septic shock’ has been widely used. Similarly, the term “sepsis, severe sepsis, and septic shock” means “sepsis without organ dysfunction, severe sepsis without shock, and severe sepsis with shock”.
- Line 43: ‘in patients with sepsis and hypotension not responding to initial fluid resuscitation’: isn’t it septic shock (lactate levels are usually increased in these circumstances)?
Answer: Thank you for your helpful comments. Of course, as you commented, lactate levels are usually increased in sepsis with hypotension not responding to initial fluid resuscitation. However, to diagnose septic shock, sepsis patients should meet all two of the diagnostic criteria 1) vasopressor requirement to maintain a MAP of 65 mmHg and 2) serum lactate level >2 mmol/L despite adequate fluid resuscitation. Therefore, patients with sepsis-induced hypotension and serum lactate level <2 mmol/L can actually exist. Those patients are diagnosed as sepsis, not septic shock. Our study included some sepsis patients who did not meet septic shock criteria but needed vasopressors to maintain optimal MAP.
- Line 62: ‘positive quick SOFA (qSOFA) score, presence or suspicion of infection, and a SOFA score of ≥2 points…’: if they have a positive qSOFA, by definition they must have a SOFA score of at least 2.
Answer: Thank you for your helpful comments. The comments “if they have a positive qSOFA, by definition they must have a SOFA score of at least 2” is not always true. For example, someone with SBP 100, alert mentality, and RR 22 is considered as a positive qSOFA. However, if the patient had no other evidence of organ failure, he could have SOFA score of 0 or 1. The details of qSOFA criteria are different from those of SOFA score. According to Sepsis-3 definitions and recent studies, qSOFA criteria is only a screening tool for sepsis with poor prognosis outside ICU. qSOFA has limited sensitivity (50-60%) for diagnosing sepsis because it is not diagnostic tool for sepsis. That means “qSOFA negative sepsis” can exist in real clinical setting. A recent study compared the clinical outcomes between “qSOFA negative sepsis” and “qSOFA positive sepsis”. The study concluded that qSOFA positive sepsis had worse clinical outcomes than qSOFA negative sepsis. Although qSOFA was developed as only a screening tool for sepsis with poor outcome, our study adopted qSOFA score as inclusion criteria. That might have resulted in selection bias (relatively higher mortality than previous studies).
- The authors should be emphasize that the AUROC is very poor in all cases (hardly 0.75).
Answer: Thank you for your helpful comments. As you commented, the prognostic value of VISmax proved to be not good, but only fair (AUC = 0.724). Generally, AUCs are classified as follows: 1) >0.9: excellent, 2) 0.8–0.9: good, 3) 0.7–0.8: fair, 4) 0.6–0.7: poor, 5) <0.6: fail. Therefore, we concluded that VISmax had only fair prognostic value comparable to SOFA score.
Reviewer 3 Report
Interesting manuscript, but since VIS is not superior to SOFA, why bother? Just use the SOFA score. I cannot see the value of this study.
Specific comments:
1. According to sepsis-3, sepsis should be defined as life-threatening organ dysfunction caused by a dysregulated host response to infection. For clinical operationalization, organ dysfunction can be represented by an increase in the Sequential [Sepsis-related] Organ Failure Assessment (SOFA) score of 2 points or more. Outside the ICU, a qSOFA>2 points. You enrolled patienst with suspected infection, and a SOFA score (or qSOFA) more than two points.
Please clarify.
2. Why did you use SOFA, and not qSOFA in the analyses?
3. As seen in Table 1, non-survivors were much more severe, and hence the increased need for vasopressor therapy.
So, is increased VISmax a prognostic marker, or merely a need for these patients?
4. Age, SOFA scores, initial lactate levels and fluids were associated with 30-day mortality in the multivariate Cox proportional hazards model.
5. VISmax is not better than SOFA on the ROC curve.
6. Conclusion: VISmax was superior to the cardiovascular component of the SOFA score and initial lactate levels and comparable to the SOFA score in its ability to predict 30-day mortality. So why not just use the SOFA score?
Author Response
Reviewer #3
Comments and Suggestions for Authors
Interesting manuscript, but since VIS is not superior to SOFA, why bother? Just use the SOFA score. I cannot see the value of this study.
Answer: Thank you for your helpful comments. VIS was not superior and comparable to SOFA in predicting mortality. We agree with your comment about the limited value of VIS. Our point is that VIS calculated in ED can be used as an alternative for outcome prediction in septic patients.
Specific comments:
- According to sepsis-3, sepsis should be defined as life-threatening organ dysfunction caused by a dysregulated host response to infection. For clinical operationalization, organ dysfunction can be represented by an increase in the Sequential [Sepsis-related] Organ Failure Assessment (SOFA) score of 2 points or more. Outside the ICU, a qSOFA>2 points. You enrolled patients with suspected infection, and a SOFA score (or qSOFA) more than two points.
Please clarify.
Answer: According to the latest Sepsis-3 definitions, qSOFA is a bedside screening for sepsis with poor prognosis outside ICU. Basically, qSOFA is only a screening tool and not a diagnostic tool. Sepsis-3 recommends the use of SOFA score to determine the presence of organ dysfunction. However, our study used both qSOFA and SOFA score as inclusion criteria. The reason is that all sepsis patients in the present study were initially enrolled by i-SMS (qSOFA alert system) in the ED of our institution. Therefore, all enrolled patients in the present study met 1) initial positive qSOFA criteria, 2) presence of infection, and 3) SOFA score of ≥2 points. That might have resulted in selection bias (relatively higher mortality than previous studies).
- Why did you use SOFA, and not qSOFA in the analyses?
Answer: Thank you for your comments. qSOFA is a bedside screening tool, not a diagnostic tool for sepsis. Although qSOFA is a quick and simple tool for screening sepsis, SOFA score is more systematic and sophisticated tool for evaluating organ dysfunction in sepsis. Therefore, we used SOFA score instead of qSOFA in the analyses.
- As seen in Table 1, non-survivors were much more severe, and hence the increased need for vasopressor therapy. So, is increased VISmax a prognostic marker, or merely a need for these patients?
Answer: Thank you for your helpful comments. As you commented, non-survivors needed significantly higher VISmax than survivors. Our study showed that VISmax was associated with short-term mortality. This results suggest that septic patients with poor outcomes needed more vasoactive and inotropic support.
4.Age, SOFA scores, initial lactate levels and fluids were associated with 30-day mortality in the multivariate Cox proportional hazards model.
Answer: Thank you for your comments. As you commented, age, SOFA score, lactate levels and fluid were associate with 30-day mortality. “Septic shock” had a positive correlation with initial lactate levels, and was finally excluded in multivariate Cox proportional hazard model using “backward LR (likelihood ratio) elimination method”. The method can remove multicollinearity.
5.VISmax is not better than SOFA on the ROC curve.
Answer: Thank you for your comments. VISmax was not superior to SOFA score on the ROC curve. VISmax proved to be comparable to SOFA score in predicting mortality. Our conclusion is that early VISmax in ED showed fair predictive value for predicting mortality.
- Conclusion: VISmax was superior to the cardiovascular component of the SOFA score and initial lactate levels and comparable to the SOFA score in its ability to predict 30-day mortality. So why not just use the SOFA score?
Answer: Thank you for your helpful comments. As you commented, prognostic value of VISmax was not superior to that of SOFA score. SOFA score is well-known predictor of organ dysfunction and clinical outcome. The point is that VISmax in ED also could be a decent alternative for predicting mortality in septic patients.
Round 2
Reviewer 1 Report
Thank you so much for answering all my questions. Some doubts have been clarified. However; I still don't know what criteria the authors have used to define a patient as a septic shock. According to table 2 there are patients who are not classified as septic shock (VIS 6–15 x 18%; VIS 16–30–24%; VIS 31–45 x 15.4%; VIS >45 x 2.7%) but they have presumably received vasopressors and inotropes and also have serum lactate levels above 2.
It would be interesting to conduct a sub-study considering only patients with Septic Shock.
Reviewer 2 Report
The authors did minimal changes in the text. It seems they did not even read the comments that could improve the manuscript. This is very disappointing.
- There is no rationale: Why is it important to define prognosis? Obviously not to limit therapy. Obviously all these patients must be admitted in the ICU, so that it has no implications for patient management. The SOFA score was introduced to describe the severity of organ dysfunction, but not to predict outcome.
- The predictive value is actually very poor (AUROC hardly above 0.75). This should be clearly recognized. The Figure 3 could be omitted. (the Figure 2 is better). The authors replied they agreed, but did not change anything in the manuscript.
- Title and main text: ‘sepsis and septic shock’ do not make sense: septic shock is a subset of sepsis. All patients with septic shock have sepsis. Septic patients may or may not be in shock. The population includes sepsis with or without shock, even though it is difficult to understand why these agents would be administered in the absence of shock.
- As the authors indicate, dopamine should not be used, epinephrine use if limited to extremely severe cases, milrinone is hardly used and the use of vasopressin is highly variable. Which agents were used in these patients? Is the max dose of norepi not an easier information about the severity of the shock state? (that is the information that people use in their communication).
Other comment:
Lines 60- 62: ‘positive quick SOFA (qSOFA) score, presence or suspicion of infection, and a SOFA score of ≥2 points…’: please correct. If the patients had a positive qSOFA, by definition they must have a SOFA score of at least 2.
Reviewer 3 Report
The authors have answered, howeverm they conclude the same as me:
Answer: Thank you for your helpful comments. VIS was not superior and comparable to SOFA in predicting mortality. We agree with your comment about the limited value of VIS. Our point is that VIS calculated in ED can be used as an alternative for outcome prediction in septic patients.
You have not justified the use of an alternative however.
I am sorry but I do not find the present study of high priority.
